# A Hierarchical Nano to Micro Scale Modelling of 3D Printed Nano-Reinforced Polylactic Acid: Micropolar Modelling and Molecular Dynamics Simulation

**DOI:** 10.3390/nano14131113

**Published:** 2024-06-28

**Authors:** AbdolMajid Rezaei, Razie Izadi, Nicholas Fantuzzi

**Affiliations:** 1Department of Structural and Geotechnical Engineering, Sapienza University of Rome, Piazzale Aldo Moro 5, 00185 Rome, Italy; abdolmajid.rezaei@uniroma1.it; 2Department of Civil, Chemical, Environmental and Materials Engineering, University of Bologna, Viale del Risorgimento 2, 40136 Bologna, Italy; nicholas.fantuzzi@unibo.it

**Keywords:** micropolar theory, FDM 3D printing, nano–micro scale modelling, molecular dynamics simulation, silver nanoparticle reinforced polylactic acid

## Abstract

Fused deposition modelling (FDM) is an additive manufacturing technique widely used for rapid prototyping. This method facilitates the creation of parts with intricate geometries, making it suitable for advanced applications in fields such as tissue engineering, aerospace, and electronics. Despite its advantages, FDM often results in the formation of voids between the deposited filaments, which can compromise mechanical properties. However, in some cases, such as the design of scaffolds for bone regeneration, increased porosity can be advantageous as it allows for better permeability. On the other hand, the introduction of nano-additives into the FDM material enhances design flexibility and can significantly improve the mechanical properties. Therefore, modelling FDM-produced components involves complexities at two different scales: nanoscales and microscales. Material deformation is primarily influenced by atomic-scale phenomena, especially with nanoscopic constituents, whereas the distribution of nano-reinforcements and FDM-induced heterogeneities lies at the microscale. This work presents multiscale modelling that bridges the nano and microscales to predict the mechanical properties of FDM-manufactured components. At the nanoscale, molecular dynamic simulations unravel the atomistic intricacies that dictate the behaviour of the base material containing nanoscopic reinforcements. Simulations are conducted on polylactic acid (PLA) and PLA reinforced with silver nanoparticles, with the properties derived from MD simulations transferred to the microscale model. At the microscale, non-classical micropolar theory is utilised, which can account for materials’ heterogeneity through internal scale parameters while avoiding direct discretization. The developed mechanical model offers a comprehensive framework for designing 3D-printed PLA nanocomposites with tailored mechanical properties.

## 1. Introduction

Additive manufacturing (AM) technology enables the creation of intricate and lightweight structures that are challenging to produce with conventional manufacturing techniques. The manufacturing industry has developed various AM methods, including stereolithography (SLA), selective laser sintering (SLS), laminated object manufacturing (LOM), solvent cast direct writing (SC-DW), and fused deposition modelling (FDM) [1].

Among these methods, FDM stands out as a particularly popular and rapidly advancing 3D printing technique. This technology offers numerous benefits, such as cost efficiency, excellent reliability, a wide selection of affordable filament materials, minimal maintenance requirements, and a relatively modest initial investment cost [2].

FDM has revolutionised the fabrication of complex geometric structures across various industries, ranging from healthcare to aerospace. In this method, a continuous filament of a thermoplastic polymer is used to print layers of materials (Figure 1a).

FDM-printed components often exhibit mechanical deficiencies attributed to void formation between filaments during the printing process [3] (Figure 1b). The presence of voids also results in anisotropic properties [4]. The formation of voids in FDM is influenced by various process parameters, such as layer thickness [5], nozzle diameter [6], extrusion temperature [7], and infill density [8].

While voids are generally regarded as defects, they can also offer advantages in certain applications. The presence of voids can help in customising specific properties such as flexibility or weight reduction [9], impact resistance [10], and thermal conductivity [11]. Moreover, voids increase the permeability of the material [12], which is beneficial for tissue engineering scaffolds [13], where nutrient diffusion is required [14,15].

**Figure 1 nanomaterials-14-01113-f001:**
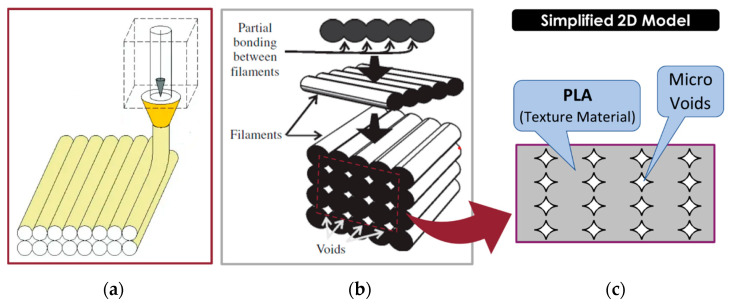
(**a**) A schematic of the FDM 3D printing process (from [16], used under the Creative Commons CC-BY license). (**b**) Formation of voids between filaments during the FDM process (from [17] ©Elsevier, used with permission under the Creative Commons CC-BY-NC-ND license). (**c**) Porous 2D model representing the cross-section of the FDM-produced component.

It is shown that the interaction of adjacent layers during deformation plays a significant role in determining the overall mechanical properties of FDM parts [18]. The porosity resulting from void formation significantly affects the structural integrity and strength of the printed parts [1,3]. Advanced techniques such as X-rays have been employed to analyse void size, shape, and distribution in FDM samples, offering insights into the effect of voids on mechanical properties [19].

The FDM process supports a diverse range of materials, including polymers, polymer matrix composites (PMC), bio-composites, polymer ceramic composites (PCC), nanocomposites, and fibre-reinforced composites (FRC) [20]. Common polymers used in FDM include Polylactic Acid (PLA), Acrylonitrile Butadiene Styrene (ABS), Polyethylene Terephthalate Glycol (PETG), Nylon, and Polyether Ether Ketone (PEEK). Each of these materials offers distinct advantages: PLA is known for its biodegradability and biocompatibility and is widely used in FDM due to its ease of printing and compatibility with various additives [18]. ABS is known for its strength and durability, PETG for its toughness and chemical resistance, Nylon for its flexibility and wear resistance, and PEEK for its exceptional mechanical properties, chemical resistance, and high-temperature performance.

Focusing on the mechanical performance of FDM-printed parts, a comparison of the tensile strength and stiffness modulus of ABS, PLA, Nylon, and PEEK polymers, based on data from various studies [1], reveals that PEEK generally offers superior mechanical and thermal properties, with PLA ranking second. However, PLA is biodegradable, easier to process and print, and significantly cheaper than PEEK.

Various reinforcement elements, such as metallic nanoparticles, carbon-based nanostructures, and both continuous and short fibres of glass, carbon, and Kevlar [1], can be integrated into the FDM process to create enhanced composites. For instance, pure PLA is limited in functionality due to its mechanical [1] and lack of antibacterial features [19]. Different nanomaterials, such as carbon nanotubes (CNTs), graphene, nano-clays, natural compounds, nanoceramics, and metallic and metallic oxide nanoparticles, are used for reinforcing PLA [21]. Each nano-additive endows unique properties to the resulting nanocomposites while also presenting certain limitations. For instance, carbon-based nanofillers such as carbon nanotubes, carbon nanofibers, and graphene significantly enhance the mechanical properties of the polymer, provided that good dispersion within the polymer matrix is ensured. However, severe concerns exist regarding the toxicity of these nano-additives, which hinder their application in biomedical and food packaging contexts [22,23]. PLA nanocomposites reinforced with metallic and metallic oxide nanoparticles, such as silver, gold, copper, silica, and titanium oxide, are commonly used as they not only improve the mechanical features but also confer additional benefits such as improved antimicrobial and antiviral properties, thermal stability, and glass transition temperature [24,25]. Among these, silver nanoparticles are of great interest as they enhance the mechanical properties of PLA, such as Young’s modulus, tensile strength, and toughness [26,27,28,29], while possessing high antimicrobial properties [30]. This makes them ideal for versatile applications in biomedical fields such as tissue engineering [31,32,33], food packaging [34], and cosmetic and hygiene products [35]. Silver nanoparticles also offer additional advantages, such as higher temperature stability and low volatility [36].

By introducing silver nanoparticles into PLA, it is possible to create composite filaments with improved mechanical properties and, at the same time, provide enhanced antimicrobial properties [6], offering a versatile solution for manufacturing medical-grade components using FDM technology [37].

Therefore, understanding the relationship between the parameters of the parent material, void characteristics, and mechanical properties is essential for optimising FDM processes and improving the overall performance of FDM-printed components. The intensity of the effect of void formation on mechanical properties and how to rectify these effects through nano-additives are not addressed thoroughly [11]. The FDM components’ internal structure is composed of polymer filaments that are partially linked and some voids (Figure 1b). Hence, the mechanical characteristics are determined by the material of the filament as well as the form and density of the voids [37,38]. By investigating the impact of material and process parameters on mechanical properties using multiscale models, researchers aim to enhance the quality and performance of FDM-manufactured components.

This study introduces a multiscale model that can predict the mechanical stiffness characteristics of components produced using FDM. Molecular dynamics (MD) simulations are used to derive the mechanical properties of the filament’s parent material at the nanoscale. MD serves as a powerful tool to unravel the atomistic intricacies that dictate the material properties of pure PLA and PLA nanocomposites. In MD simulations, Newton’s equations of motion are employed to calculate the interactions between atoms and molecules. This enables the detailed analysis of atomistic events and deformation mechanisms that dictate the overall mechanical behaviour of the material. At the microlevel, the properties of the filaments are analysed in conjunction with the voids to determine the overall mechanical characteristics of the component. Nevertheless, performing direct modelling and discretization of such media at the microlevel, which includes microstructural voids, might result in complex and burdensome calculations. In this study, non-classical continuum theories were utilised because the traditional continuum theory is unable to consider the internal structure and lacks the ability to retain information about it. The non-classical theories are capable of preserving the memory of the internal structure at a microscopic level [39]. Specifically, the micropolar continuum is employed at the second scale for the homogenisation of the heterogeneous structure. The Cosserat continuum or elastic micropolar theory [40] has been effectively applied in various applications [41,42] to characterise different kinds of heterogeneous materials [43], including porous [15,44,45,46,47,48] and cellular structures [49], composites [50,51], lattices [52], foams [53,54,55], and nanostructures [39,56].

In MD simulation, unit cells consisting of PLA and PLA reinforced with different weight fractions of silver nanoparticles are constructed, and the elastic properties are determined by applying uniaxial tension/compression and pure shear strains. At the microscale, the mechanical properties of the continuum are determined by considering the strain energy equivalence of a porous two-dimensional geometry representing the cross-section of the FDM final part (shown in Figure 1c) with the corresponding micropolar homogenised model under different loadings. Numerical simulations are used to explore the impact of different features, such as void patterns and sizes, that arise from printing parameters, on the material properties. The constructed mechanical model offers a framework for the creation of 3D-printed PLA components with tailored mechanical characteristics based on the printing conditions.

The remainder of this paper is structured as follows: Section 2 is devoted to the details of the molecular dynamics simulation conducted to find the mechanical properties of pure PLA and PLA nanocomposites as the parent material of the FDM filaments. Section 3 presents the homogenisation procedure for modelling the FDM component with void patterns and how to extract the micropolar material parameters. It also describes the parametrisation of the geometrical features corresponding to the FDM process parameters. In Section 4, the developed model is used to conduct a parametric study of the effect of FDM void features and the properties of the parent material on the mechanical properties. Finally, Section 5 summarises the key findings and outlines future research directions in the fields of homogenisation and multiscale approaches to find mechanical properties of FDM-produced components resulted from 3D printing.

## 2. Molecular Dynamics Simulation of PLA Reinforced by Nano Silver Particles

### 2.1. Materials

In this section, all-atom molecular dynamics simulations are implemented to find the mechanical properties of pure PLA and PLA nanocomposites that are reinforced with different weight fractions of silver nanoparticles. The simulations are conducted in Materials Studio 2019 software. Condensed Phase Optimised Molecular Potential for Atomistic Simulation Studies (COMPASS) is employed to describe the atomic interactions. The COMPASS force field has proven to be effective and accurate in handling polymer subjects in condensed matter [57]. Figure 2a shows the chemical structure of PLA with a molecular formula of ${{(C}_{3}H_{4}O_{2})}_{n}$ [58], and Figure 2b shows the 20-monomer polymer chains utilised in the present study. PLA possesses several stereoisomers, including poly(L-lactide) (PLLA), poly(D-lactide) (PDLA), and poly(DL-lactide) (PDLLA), due to the chirality of the lactic acid molecule [59,60]. Based on Figure 2, the polymer studied in this research is PDLA because of the methyl group’s location in relation to the chain backbone [61,62].

Silver nanoparticles (Ag-NP) with a diameter of 1.0 nm, as shown in Figure 3a, are considered as nano-reinforcements. To construct the molecular models of the nanocomposites, the nano-silver particle is placed in the centre of a cubic representative volume element (RVE). Subsequently, the PLA chains are evenly distributed throughout the RVE without overlapping. The RVE’s initial configuration is produced using the Amorphous Builder Module, a feature in Materials Studio software [63] that utilises a Monte Carlo method to assemble molecules. Figure 3 displays the RVEs containing 21.8% and 6.6% weight fractions of Ag-NP.

### 2.2. Relaxation

The energy minimisation process is controlled by a smart algorithm that combines steepest descent, adjusted basis Newton–Raphson, and quasi-Newton method in a cascading manner. Secondly, in order to enhance the movement of polymer chains and expedite the process of reaching equilibrium, the temperature of the RVE is raised and kept at 500 K, through an NVT ensemble. This temperature is higher than the glass transition temperature (Tg) of PLA, which is predicted to be between 323 K and 353 K [64]. The system is subsequently cooled down to a temperature of 298 K over a time period of 50 picoseconds using the NPT ensemble. The Nose–Hoover thermostat and barostat are employed to regulate the temperature and pressure using a stochastic time integrator with a time step of 1 femtosecond (fs) [65]. Periodic boundary conditions are enforced to replicate the cells in three directions. The cutoff radius for non-bonded interactions in all simulations is set to 1.5 nm.

### 2.3. Elastic Constants

The elastic constants of the nanocomposite RVEs are then calculated using the constant strain approach, as proposed by Theodorou and Suter [66]. This method analyses the static deformation of the RVE under uniaxial tension/compression and pure shear strains. As detailed in [67], the Voigt–Reuss–Hill approach is used to obtain Young’s modulus and Poisson’s ratio. This approach provides a reliable approximation for calculating the effective elastic constants. The obtained results for pure PLA and PLA reinforced nanocomposites with filler weight fractions of 6.5% and 21.8% are reported in Table 1. The obtained properties in each RVE are averaged between three different initial configurations.

As shown in Table 1, increasing the weight fraction of nano-silvers significantly increases the Young’s modulus and slightly decreases the Poisson’s ratio. This trend implies that by adding nano-reinforcement, the material becomes stiffer and less deformable, exhibiting more brittleness.

This effect can be related to the formation of an interphase region, which is a crystalized layer of polymer around the nanofiller with increased density and improved mechanical parameters. It is believed that the interphase region has the main role in the stiffening mechanism of nanofillers in nanocomposites [68]. By studying the density distribution function of the polymer around the nanoparticle as shown in Figure 4, the presence of an interphase layer, characterised by a localised higher density, is confirmed.

## 3. Homogenisation

A possible approach for modelling the FDM component with microstructures (void patterns) is to employ a multiscale approach. This involves using a homogenised medium to represent the heterogeneous structure. The primary assumption for determining the constitutive parameters of such equivalent model is that the strain energy stored in the heterogeneous structure with voids at the micro-level, is equal to that of the homogenous equivalent continuum at the macro-level. The current study employs the classical Cauchy continuum at the micro-level and the micropolar continuum at the macro-level, as depicted in Figure 5.

### 3.1. Planar Micropolar Theory

In the micropolar theory, the continuum’s material particles possess an extra degree of freedom, called microrotation, in addition to the standard displacement field. The linearized kinematic equations given below describe a micropolar continuum [39,69]:(1)$$E_{ij}=U_{i,j}+e_{ijk}\Phi_{k}\;M_{kj}=\Phi_{k,j}$$
where $U_{i}$ and $\Phi_{k}$ denote the components of displacement and micro-rotation vectors, whereas $E_{ij}$ and $M_{kj}$ represent the components of strain and curvature tensors, respectively. $e_{ijk}$ is the typical third order permutation symbol.

If the effects of body forces ($P_{i}$) and body couples ($Q_{k}$) are taken into account, the equilibrium equations can be expressed as follows:(2)$$\Sigma_{ij,j}+P_{i}=0\;M_{kj,j}-e_{ijk}\Sigma_{ij}+Q_{k}=0$$
where $\Sigma_{ij}$ and $M_{kj}$ represent the components of the non-symmetric stress and couple-stress tensors, respectively.

Considering the two-dimensional formulation of the micropolar theory, there are two translational displacements (U,V) and one rotational displacement (Φ). Therefore, the generalised displacement vector can be expressed as follows:(3)$$U^{T}=\left\{\begin{array}{ccc} U & V & \Phi \end{array}\right\}$$
and the strain vector is represented as follows:(4)$$E^{T}=\left\{\begin{array}{cccccc} E_{11} & E_{22} & E_{12} & E_{21} & K_{1} & K_{2} \end{array}\right\}$$
where $E_{11},E_{22},E_{12},E_{21}$ are the in-plane normal and shear strains, and $K_{1},K_{2}$ are the micropolar curvatures.

The stress vector is also represented as follows:(5)$$\Sigma^{T}=\left\{\begin{array}{ccc} \begin{array}{cccc} \Sigma_{11} & \Sigma_{22} & \Sigma_{12} & \Sigma_{21} \end{array} & M_{1} & M_{2} \end{array}\right\}$$
where $\Sigma_{ij}$ (*i*, *j* = 1, 2) indicates the normal (*i* = *j*) and shear (*i* ≠ *j*) stress components, whereas $M_{1}$, $M_{2}$ represent the planar micro-couples.

The general anisotropic constitutive equations for the micropolar continua can be formulated as:(6)Σ=CE.

The constitutive stiffness matrix, denoted as **C**, is symmetrical due to the hyper elastic nature of the considered material [70].

The geometries considered here for the 2D model of the FDM-produced component (such as the one shown in Figure 5) are orthotropic. The constitutive equations can be expressed using Voigt’s notation as follows:(7)$$\left\{\begin{array}{c} \begin{array}{c} \Sigma_{11}\\ \Sigma_{22} \end{array}\\ \begin{array}{c} \Sigma_{12}\\ \Sigma_{21} \end{array}\\ \begin{array}{c} M_{1}\\ M_{2} \end{array} \end{array}\right\}=\left[\begin{array}{cccccc} A_{1111} & A_{1122} & 0 & 0 & 0 & 0\\ A_{1122} & A_{2222} & 0 & 0 & 0 & 0\\ 0 & 0 & A_{1212} & A_{1221} & 0 & 0\\ 0 & 0 & A_{1221} & A_{2121} & 0 & 0\\ 0 & 0 & 0 & 0 & D_{11} & 0\\ 0 & 0 & 0 & 0 & 0 & D_{22} \end{array}\right]\left\{\begin{array}{c} E_{11}\\ \begin{array}{c} E_{22}\\ \begin{array}{c} E_{12}\\ E_{21} \end{array}\\ K_{1} \end{array}\\ K_{2} \end{array}\right\}$$

Therefore, the stiffness matrix in the current work contains eight independent micropolar material parameters. These parameters are as follows:$$A_{1111},A_{2222},A_{1122},A_{1212},A_{2121},A_{1221},D_{11},D_{22}.$$

In addition, we employ an alternative description of the micropolar shear deformation elements to explicitly identify the initial three components of strain as the in-plane Cauchy components ($E_{11},E_{22},E_{12}^{SYM}$). By defining the following:(8)$$E_{12}^{SYM}=\frac{E_{12}+E_{21}}{2},\;\;\;\;\Theta =\frac{E_{12}-E_{21}}{2}$$

The micropolar strain E can be expressed as follows:(9)$$E^{T}=\left\{\begin{array}{cccccc} E_{11} & E_{22} & E_{12}^{SYM} & \Theta & K_{1} & K_{2} \end{array}\right\}$$

In which $E_{12}^{SYM}$ is the symmetric part of the shear strain components, whereas Θ denotes the antisymmetric part.

The work conjugates for these two shear strain measures are found to be as follows:(10)$$\begin{array}{l} \Sigma_{12}^{SYM}=\frac{1}{2}\left(\Sigma_{12}+\Sigma_{21}\right)\\ \Sigma_{12}^{ASM}=\frac{1}{2}\left(\Sigma_{12}-\Sigma_{21}\right) \end{array}$$
where $\Sigma_{12}^{SYM}$ and $\Sigma_{12}^{ASM}$ are the symmetric and antisymmetric components of the shear stress.

Finally, Equation (7) can be expressed in terms of $E_{12}^{SYM}$ and Θ as follows:(11)$$\left\{\begin{array}{c} \begin{array}{c} \Sigma_{11}\\ \Sigma_{22} \end{array}\\ \begin{array}{c} \Sigma_{12}\\ \Sigma_{21} \end{array}\\ \begin{array}{c} M_{1}\\ M_{2} \end{array} \end{array}\right\}=\left[\begin{array}{cccccc} A_{1111} & A_{1122} & 0 & 0 & 0 & 0\\ A_{1122} & A_{2222} & 0 & 0 & 0 & 0\\ 0 & 0 & A_{1212}^{\prime } & A_{1221}^{\prime } & 0 & 0\\ 0 & 0 & A_{1221}^{\prime } & A_{2121}^{\prime } & 0 & 0\\ 0 & 0 & 0 & 0 & D_{11} & 0\\ 0 & 0 & 0 & 0 & 0 & D_{22} \end{array}\right]\left\{\begin{array}{c} E_{11}\\ \begin{array}{c} E_{22}\\ \begin{array}{c} E_{12}^{SYM}\\ \Theta \end{array}\\ K_{1} \end{array}\\ K_{2} \end{array}\right\}$$
where it can be shown that:(12)$$\begin{array}{l} A_{1212}^{\prime }=\frac{A_{1212}+A_{2121}+2A_{1221}}{4},\\ A_{2121}^{\prime }=\frac{A_{1212}+A_{2121}-2A_{1221}}{4},\\ A_{1221}^{\prime }=\frac{A_{1212}-A_{2121}}{4} \end{array}$$

### 3.2. Identification of Equivalent Micropolar Material Parameters

Due to the distinct degrees of freedom employed by micropolar and Cauchy continua, a kinematic map is necessary to provide a connection between these two levels of description. In this study, we utilised the mapping method described by Forest and Sab [43] for an RVE in the shape of a square and modified it for a rectangular RVE in [71] (see Figure 6).
(13)$$\begin{array}{l} u=E_{11}x+E_{12}^{SYM}y-\frac{K_{2}}{2}y^{2}-K_{1}xy+\frac{5(b^{2}+h^{2})}{b^{4}}\left[E_{12}^{SYM}\left(\frac{h^{2}-b^{2}}{b^{2}+h^{2}}\right)+\Theta \right](y^{3}-3\frac{h^{2}}{b^{2}}x^{2}y)\;\\ v=E_{12}^{SYM}x+E_{22}y+\frac{K_{1}}{2}x^{2}+K_{2}xy-\frac{5(b^{2}+h^{2})}{b^{4}}\left[E_{12}^{SYM}\left(\frac{h^{2}-b^{2}}{b^{2}+h^{2}}\right)+\Theta \right](\frac{h^{2}}{b^{2}}x^{3}-3xy^{2})\; \end{array}$$

Equation (13) defines the estimated microscopic displacement field within the RVE (u,v) as a function of the macroscopic strains ($E_{11},E_{22},E_{12}^{SYM},\Theta ,K_{1},K_{2}$) at the material point on the macro-level.

Once the kinematic map is determined, we proceed to discover the micropolar material parameters in Equation (7). To do this, we use the finite element method (FEM) [15] to compute the response of the model with voids under different loadings, as shown in Figure 7. For each scenario, the micropolar material parameters are determined in such a way that the analogous material retains the same amount of strain energy when exposed to the same loading conditions. This equivalence means the following:(14)$$U_{FEM}=U_{Micropolar}$$
where $U_{Micropolar}$ denotes the strain energy of the equivalent micropolar continuum. This value is determined analytically by the following relation:(15)$$\begin{array}{l} U_{Micropolar}=\frac{1}{2}\left[E_{11}\Sigma_{11}+E_{22}\Sigma_{22}+E_{12}^{SYM}\Sigma_{12}^{SYM}+\Theta \Sigma_{12}^{ASM}+K_{1}M_{1}+K_{2}M_{2}\right]\\ \;\;\;\;\;\;\;\;\;\;\;\;\;\;\;\;\;\;\;\;=\frac{1}{2}[E_{11}(A_{1111}E_{11}+A_{1122}E_{22})\;+E_{22}(A_{1122}E_{11}+A_{1111}E_{22})\;+\\ \;\;\;\;\;\;\;\;\;\;\;\;\;\;\;\;\;\;\;\;\;\;\;\;\;\;\;E_{12}^{SYM}(A_{1212}^{\prime }E_{12}^{SYM}+A_{1221}^{\prime }\Theta )+\Theta (A_{1221}^{\prime }E_{12}^{SYM}+A_{2121}^{\prime }\Theta )\;+\\ \;\;\;\;\;\;\;\;\;\;\;\;\;\;\;\;\;\;\;\;\;\;\;\;\;\;K_{1}(D_{11}K_{1})+K_{2}(D_{22}K_{2})] \end{array}$$

And $U_{FEM}$ is the strain energy stored in the detailed structure with voids resulted from the FEM simulations and is extracted directly from FE software, COMSOL Multiphysics 6.0:(16)$$U_{FEM}=\frac{1}{2}\int_{RVE}\sigma_{ij}\varepsilon_{ij}dV$$
where $\sigma_{ij}$ and $\varepsilon_{ij}$ represent stress and strain values.

The different loading scenarios, referred to as FEM tests, and the related material properties acquired from each test are outlined in Figure 7.

The boundary conditions for each test are determined by utilising the micro-field descriptions of u and v, which are expressed in terms of macro-field strain measures as shown in Equation (13).

#### 3.2.1. Test 1: Uniaxial Extension Test to Find $A_{1111}$

The model is subjected to a homogeneous strain $E_{11}=1$, resulting in the following boundary conditions:(17)$$\begin{array}{l} u=E_{11}x\\ v=0 \end{array}$$

And the elastic strain energy equality from Equation (14) yields the following:(18)$$\frac{1}{2}[E_{11}(A_{1111}E_{11})]=U_{FEM,1}$$

Which gives the following:(19)$$A_{1111}=2U_{FEM,1}$$

#### 3.2.2. Test 2: Uniaxial Extension Test to Find $A_{2222}$

Similar to Test 1, by applying a uniform strain $E_{22}=1$ and the following boundary conditions:(20)$$\begin{array}{l} u=0\\ v=E_{22}y \end{array}$$

And the equivalence of the elastic strain energy density results in the following:(21)$$A_{2222}=2U_{FEM,2}$$

#### 3.2.3. Test 3: Biaxial Extension Test to Find $A_{1122}$

The RVE can be subjected to bi-axial homogeneous strains, $E_{11}=E_{22}=1$, by applying the appropriate boundary conditions:(22)$$\begin{array}{l} u=x\\ v=y \end{array}$$

And the equivalence of the elastic strain energy density leads to the following:(23)$$\frac{1}{2}[E_{11}(A_{1111}E_{11}+A_{1122}E_{22})\;+E_{22}(A_{1122}E_{11}+A_{2222}E_{22})\;]=U_{FEM,3}$$

Giving the following:(24)$$A_{1122}=U_{FEM,3}-\frac{A_{1111}+A_{2222}}{2}$$

#### 3.2.4. Test 4: Symmetric Shear Deformation Test to Find $A_{1212}^{\prime }$

To apply uniform shear strain $E_{12}^{SYM}=1$ to the equivalent micropolar media, the corresponding boundary conditions for the micromodel should be as follows:(25)$$\begin{array}{l} u=y+\frac{5(h^{2}-b^{2})}{b^{4}}(y^{3}-3\frac{h^{2}}{b^{2}}x^{2}y)\;\\ v=x-\frac{5(h^{2}-b^{2})}{b^{4}}(\frac{h^{2}}{b^{2}}x^{3}-3xy^{2})\; \end{array}$$

Which results in the following elastic strain energy equivalence:(26)$$\frac{1}{2}[E_{12}^{SYM}(A_{1212}^{\prime }E_{12}^{SYM})]=U_{FEM,4}$$

Giving the following:(27)$$A_{1212}^{\prime }=2U_{FEM,4}$$

#### 3.2.5. Test 5: Rotational Deformation Test to Find $A_{2121}^{\prime }$

In order to subject the equivalent micropolar media to uniform rotational deformation, Θ=1, the relevant boundary conditions should be as follows:(28)$$\begin{array}{l} u=+\frac{5(b^{2}+h^{2})}{b^{4}}(y^{3}-3\frac{h^{2}}{b^{2}}x^{2}y)\;\\ v=-\frac{5(b^{2}+h^{2})}{b^{4}}(\frac{h^{2}}{b^{2}}x^{3}-3xy^{2})\; \end{array}$$
and the equivalence of strain energy density gives the following:(29)$$\frac{1}{2}[\Theta (A_{2121}^{\prime }\Theta )]=U_{FEM,5}$$
giving the following:(30)$$A_{2121}^{\prime }=2U_{FEM,5}$$

#### 3.2.6. Test 6: Shear Deformation Test to Find $A_{1221}^{\prime }$

By applying the boundary conditions described in Test 4 and Test 5 concurrently, the equivalence of strain energy density results in the following:(31)$$\frac{1}{2}\left[E_{12}^{SYM}(A_{1212}^{\prime }E_{12}^{SYM}+A_{1221}^{\prime }\Theta )+\Theta (A_{1221}^{\prime }E_{12}^{SYM}+A_{2121}^{\prime }\Theta )\;\right]=U_{FEM,6}$$
giving following:(32)$$A_{1221}^{\prime }=U_{FEM,6}-\frac{A_{1212}^{\prime }+A_{2121}^{\prime }}{2}$$

Using the results of the symmetric shear (Equation (27)), rotational deformation (Equation (30)), and shear tests (Equation (32)) together, the material parameters $A_{1212}$, $A_{1221}$, and $A_{1221}$ can be found from Equation (12) as follows:(33)$$\begin{array}{l} A_{1212}=A_{1212}^{\prime }+A_{2121}^{\prime }+2A_{1221}^{\prime }=2U_{FEM,6}\\ A_{2121}=A_{1212}^{\prime }+A_{2121}^{\prime }-2A_{1221}^{\prime }=4\left(U_{FEM,4}+U_{FEM,5}\right)-2U_{FEM,6}\\ A_{1221}=A_{1212}^{\prime }-A_{2121}^{\prime }\;\;\;\;\;\;\;\;\;\;\;\;\;\;\;\;=2\left(U_{FEM,4}-U_{FEM,5}\right) \end{array}$$

#### 3.2.7. Test 7: Uniform Bending Test to Find $D_{11}$

Finally, when uniform curvature $K_{1}=1$ is applied to the equivalent micropolar media, the corresponding boundary conditions will be as follows:(34)$$\begin{array}{l} u=-xy\\ v=\frac{x^{2}}{2} \end{array}$$

Resulting in the following elastic strain energy equivalence:(35)$$\frac{1}{2}[K_{1}(D_{11}K_{1})]=U_{FEM,7}$$

Which leads to the following:(36)$$D_{11}=2U_{FEM,7}$$

#### 3.2.8. Test 8: Uniform Bending Test to Find $D_{22}$

Similar to Test 7, by applying a uniform curvature $K_{2}=1$ and the following boundary conditions:(37)$$\begin{array}{l} u=-\frac{y^{2}}{2}\;\\ v=xy \end{array}$$

The equivalence of the elastic strain energy density results in the following:(38)$$D_{22}=2U_{FEM,8}$$

### 3.3. Implementation of the Finite Element Method

The finite element method was implemented in COMSOL Multiphysics, with the porous structure discretized using first-order (linear) triangular elements. Considering the plate’s thickness was substantial compared to the in-plane microstructure, a plane strain formulation for linear elastic media was adopted.

### 3.4. Parametrization of the Microstructure Resulted from the FDM Process

#### 3.4.1. Modelling of Filaments

While the material is extruded through a circular nozzle, its cross-section becomes deformed during deposition due to interaction with the platform or the underlying deposited layer, and the pressure from the moving nozzle (Figure 8a). The filament’s cross-section, after deposition, can be modelled as a combination of two semicircles and a rectangle. This parametrization is also used by well-known slicing programmes like Slic3r [72] and PrusaSlicer [73] for generating the extrusion toolpath [74]. In this context, the cross-sectional geometry is primarily defined by the layer height $H_{0}$ and the flat width, $W_{0}$, as shown in Figure 8b.

#### 3.4.2. Modelling of Bond Formation between Filaments

Figure 9a describes the sintering process occurring during FDM manufacturing. Based on the original work by [75] and followed by [37], the sintering process is idealised in the case of two isolated filaments. During sintering, it is assumed that the layer height, $H_{0}$ and length of each filament, $L_{0}$ (Figure 8b) remain constant. Initially (*t* = 0), the cross-sectional geometry consists of a rectangle with an initial width of $W_{0}$ and two semicircles with radii of $H_{0}/2$ with only one contact point between the two filaments. As the process progresses, two semi-circles in contact move closer to one another, eventually overlapping to some extent and forming an intersection, where θ(t) is the intersection angle (Figure 9b). Keeping mass constant during sintering is essential for having a proper description of the problem. Since the length and height of the layers are considered constants, we may assume that the width of the rectangle, W(t), changes with time.

By assuming a constant density for two successive filaments and using the principle of conservation of mass, we can determine the evolution of W(t) as a function of θ(t) [37]:(39)$$W(t)=W_{0}+\frac{H_{0}\theta (t)}{4}-\frac{H_{0}\sin\left(2\theta (t)\right)}{8}$$

The angle of intersection, θ, can be an indirect measure of the quality of bonding between the two adjacent filaments, as all the significant FDM process parameters are reflected in this value. For instance, based on the procedure followed in [37,75], the evolution of θ(t) can be numerically modelled as follows:(40)$$\frac{d\theta (t)}{dt}=\frac{\Gamma \left(T_{r}(x,t)\right)H_{0}}{\eta \left(T_{r}(x,t)\right)}\frac{2\cos^{2}\left(\theta (t)\right)}{\left(\pi {\left(\frac{H_{0}}{2}\right)}^{2}+H_{0}W_{0}\right){\left(\frac{-\frac{H_{0}}{2}\sin\left(\theta (t)\right)+\frac{H_{0}}{8}-\frac{H_{0}\cos\left(2\theta (t)\right)}{8}}{\frac{H_{0}}{2}+\frac{W_{0}}{2}}\right)}^{2}}$$

$T_{r}(x,t)$ represents the temperature of the filament as it changes over time. This parameter can be estimated by the use of a thermal evolution model, which determines the temperature at the moment when two filaments come into contact during the 3D printing process:(41)$$T_{r}(x,t)=T_{extr}e^{\beta \left(t-t_{r}(x)\right)}+\frac{1-e^{\beta \left(t-t_{r}(x)\right)}}{K_{cnv}+K_{cnd}}\left(\sum_{i=1}^{n}a_{ri}h_{i}\lambda_{i}T_{ri}(t)+K_{cnv}T_{E}+a_{r\mathrm{supp}}\lambda_{\mathrm{supp}}h_{\mathrm{supp}}T_{\mathrm{supp}}\right)$$

A detailed description of these equations and their derivation can be found in [20]. Briefly, $\Gamma \left(T_{r}(x,t)\right)$ and $\eta \left(T_{r}(x,t)\right)$ are the temperature-dependent coefficients of surface tension and material viscosity, respectively. The term $K_{cnv}$ defines heat transfer by convection with the environment per unit temperature, while $K_{cnd}$ defines conduction between adjacent filaments or support per unit temperature. β is defined as follows:(42)$$\beta =\frac{-P\left(K_{cnv}+K_{cnd}\right)}{\rho CA}$$

The variables used in Equations (40)–(42) as well as other model variables, are summarised and defined in Table 2 (see for further detail [37]).

As can be seen from the described model, all the FDM main process parameters play roles in determining the value of θ(t). These process parameters include printing parameters (layer thickness, nozzle size, extrusion temperature, printing speed), material properties (density, thermal conductivity, viscosity, surface tension coefficient), and printing environment (room temperature, heat convection coefficient).

In the current work, we will consider θ as the indirect input parameter that can describe the final idealised microstructure of the FDM-produced component based on the 3D printing process parameters. The overall methodology is illustrated in Figure 10.

## 4. Numerical Results and Discussion

An FDM-produced component with the size and process parameters shown in Table 3 is studied in this section, and the effect of bonding quality and properties of the parent material on the mechanical parameters is investigated.

In Figure 11, the obtained equivalent micropolar parameters for the intersection angles ranging from θ=10° to θ=30° are presented for PLA and also two nanocomposites of PLA with a 6.5% Ag reinforcement and a 21.8% Ag reinforcement.

It should be noted that the obtained micropolar parameters can further be used to study the mechanical performance of the final FDM-produced component in the framework of the micropolar continuum theory, as studied in [15].

The micropolar model implemented in this work is orthotropic to be consistent with the physical symmetries in the FDM components. The constitutive equation contains eight independent micropolar material parameters which are $A_{1111},A_{2222},A_{1122},A_{1212},A_{2121},A_{1221},D_{11},D_{22}$, whereas the micropolar isotropic model has only four independent material parameters. As shown in Figure 11, the developed framework can account for the anisotropy in mechanical properties and capture the different material parameters in the direction of filament layer (direction 1 in Figure 12), where filament bonding occurs, and the perpendicular direction (direction 2 in Figure 12) or the direction of printing layers. The material parameters of the first direction are predicted to be higher than the second direction’s which is in line with the physics of the component.

### 4.1. Effect of Filament Bonding and Intersection Angle

According to Figure 11, by increasing the intersection angle, all the stiffness parameters increase monotonically. This observation is consistent with the physics of the model, since higher values of θ show improved quality of filament bonding and therefore a higher quality of the FDM component in terms of mechanical properties.

However, the impact of the change in intersection angle varies for different equivalent material parameters. This indicates that the improved bonding has a different effect on mechanical properties depending on the material orientations of the FDM-printed sample (see Figure 12).

The change in θ highly influences the mechanical parameters in the direction of each filament layer (direction 1), i.e., $A_{1111},D_{11}$. For instance, the change in $A_{1111}$ due to the increase in θ from $\theta =10^{\circ }$ to $\theta =30^{\circ }$ is about 41% (Figure 13a) for pure PLA filaments.

On the other hand, there is a slight change for parameters in the perpendicular direction (direction 2), i.e., $A_{2222},D_{22}$. For $A_{2222}$, the increase in θ from $\theta =10^{\circ }$ to $\theta =30^{\circ }$ changes this parameter by only 7% (Figure 13b) for pure PLA filaments.

Also, the change of parameters related to shear terms, i.e., $A_{1212},A_{2121},A_{1221}$, is moderate. For instance, the change in $A_{1212}$ and $A_{2121}$ due to the increase in θ from $\theta =10^{\circ }$ to $\theta =30^{\circ }$ is approximately 17% (Figure 13c,d) for pure PLA filaments.

Therefore, it is shown that in addition to the different parameters in each direction, the effect of the filament’s bonding quality, as reflected in θ, on the material parameters differs between directions. The model predicts a more pronounced influence of the filament’s bonding quality on the first direction compared to the second.

### 4.2. Effect of Silver Nanoparticles

According to Figure 11, reinforcing PLA with nanoparticles greatly enhances all the mechanical parameters of the FDM-produced part at each intersection angle and porosity. This observation is of value in designing biomedical implants produced from biodegradable PLA. In these applications, usually a specific level of porosity is required to ensure permeability, while higher stiffness than pure polymer is needed to ensure load-bearing capacity. As mentioned earlier, the introduction of silver nanoparticles to PLA will also endow antibacterial properties to the final component.

Focusing on Figure 11 allows us to better understand the interaction between the nanoscale and microscale phenomena. As detailed in Table 1, increasing the weight fraction of nano-silver particles at the nanoscale leads to a significant increase in the Young’s modulus and elastic stiffness of the resulting filaments at the microscale, while slightly decreasing the Poisson’s ratio, resulting in reduced deformability of each filament. These changes in the elastic properties of each filament result in higher stiffness coefficients in all directions at the microscale. For instance, for the intersection angle of $\theta =30^{\circ }$, adding 6.5% of silver nanoparticles to PLA will increase $A_{1111},\;A_{1122},A_{2222},\;A_{1212},A_{2121},A_{1221},\;D_{11}$ and $D_{22}$, up to 46%, 46%, 38%, 49%, 49%, 49%, 47%, and 47%, respectively. Also, the introduction of a 21.5% weight fraction of Ag-NP to PLA will alter these parameters by 174%, 174%, 156%, 181%, 181%, 181%, 176%, and 176%, respectively.

It can also be seen from Figure 11 that the influence of bonding parameters becomes more prominent in the case of PLA nanocomposites compared to pure PLA. This highlights that controlling the process parameters to customise mechanical properties is more crucial when dealing with nanocomposites.

## 5. Conclusions

In the current work, nano–micro scale modelling is adopted to predict the mechanical properties of FDM-produced components with inherent microstructures. At the nanoscale, using MD simulations, the mechanical properties of the filament parent material are extracted for pure PLA and PLA nanocomposites reinforced with silver nanoparticles. At the microscale, a non-classical micropolar continuum was proposed to model porous microstructure resulting from void formation during the FDM 3D printing process.

The main findings are summarised as follows:By increasing the intersection angle, all the stiffness parameters are improved, consistent with the physical nature of the component.The influence of the change in intersection angle on mechanical properties varies for different material orientations of the FDM-printed sample.The change in θ highly influences the mechanical parameters in the direction of each filament layer, while its impact in the direction of printing layers is minimal, and an average impact is found for the shear-related parameters.The introduction of silver nanoparticles to pure PLA can significantly improve the stiffness parameters of the final product at the same porosity. This is desirable in designing biomedical implants where a specific level of porosity is required to ensure permeability.The influence of bonding parameters on mechanical properties becomes more prominent in the case of PLA nanocomposites compared to pure PLA.

In future works, using the developed methodology, FDM printing with a non-matching void layout (Figure 14a) or with the presence of an air gap between layers (Figure 14b) can be studied.

Moreover, the homogenised micropolar parameters can be used to study the mechanical performance of the final FDM produced component in the framework of micropolar continuum theory.

## Figures and Tables

**Figure 2 nanomaterials-14-01113-f002:**
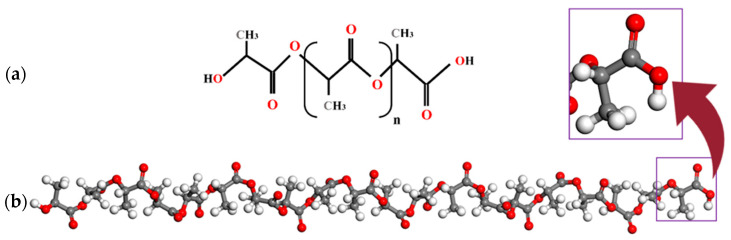
(**a**) Chemical structure of PLA (**b**) The PLA chain consists of 20 monomers. The red, grey and white colours refer to oxygen, carbon and hydrogen atoms, respectively.

**Figure 3 nanomaterials-14-01113-f003:**
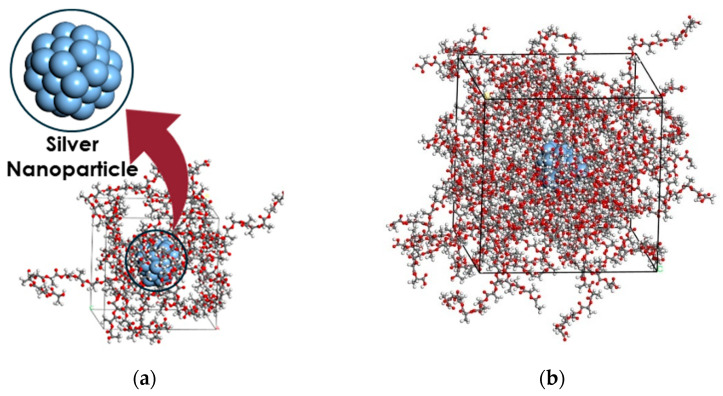
PLA reinforced by silver nanoparticles with different weight fractions ((**a**) 21.8%, (**b**) 6.50%). The blue, red, grey and white colours refer to silver, oxygen, carbon and hydrogen atoms, respectively.

**Figure 4 nanomaterials-14-01113-f004:**
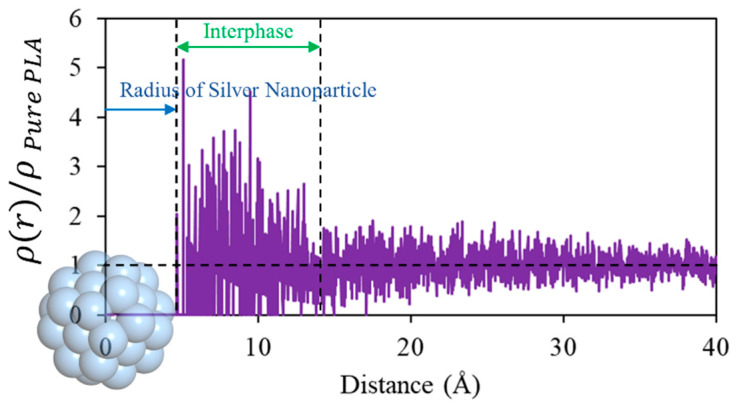
Distribution of normalised polymer density around silver nanoparticle.

**Figure 5 nanomaterials-14-01113-f005:**
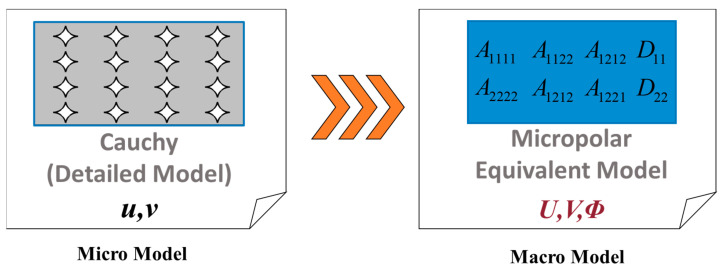
The homogenisation procedure from the classical Cauchy continuum at the micro-level to the micropolar continuum at the macro-level.

**Figure 6 nanomaterials-14-01113-f006:**
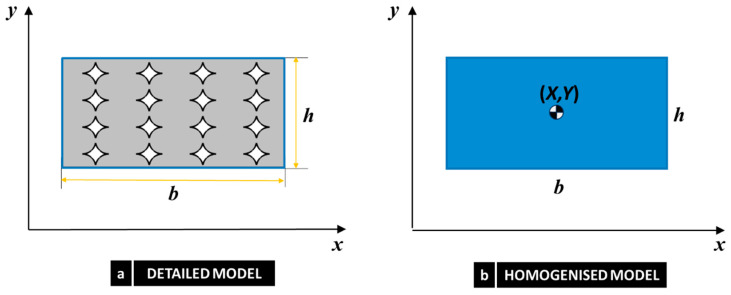
RVE Geometry: (**a**) Micro (detailed) model and (**b**) macro (homogenised equivalent micropolar) model.

**Figure 7 nanomaterials-14-01113-f007:**
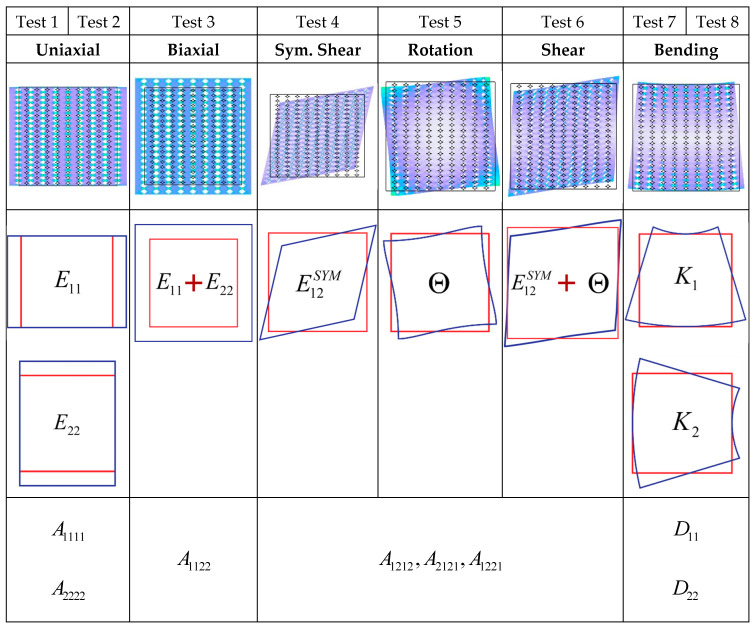
Designed FEM tests for finding micropolar material parameters. The red frames indicate the undeformed states, and the blue ones indicate the deformed configurations.

**Figure 8 nanomaterials-14-01113-f008:**
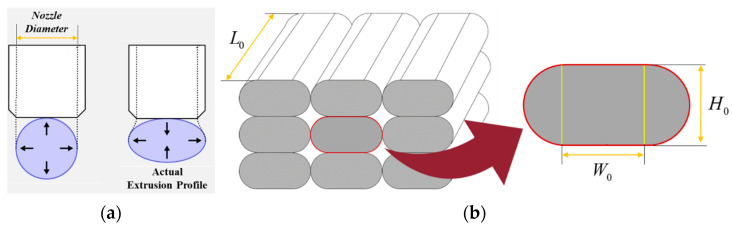
(**a**) Theoretical and actual extrusion profile in the FDM process (**b**) The approximated cross-section of a filament after deposition.

**Figure 9 nanomaterials-14-01113-f009:**
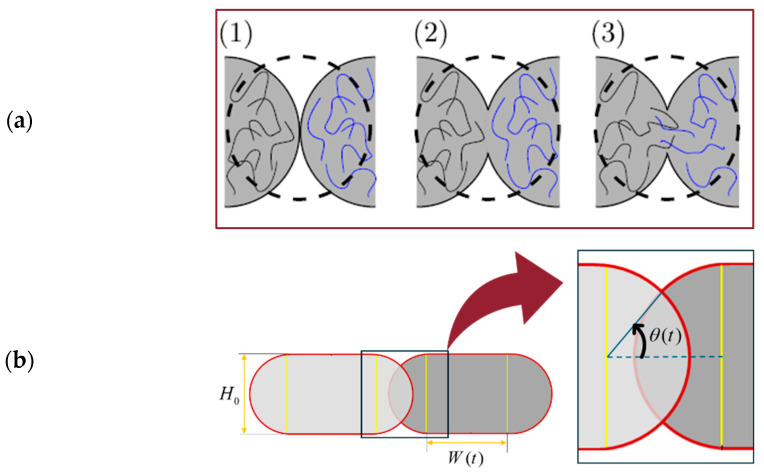
(**a**) Bond formation process through sintering: (1) surface contact; (2) neck growth; (3) neck growth and molecular diffusion at the interface (from [37] ©Elsevier, used with permission under the Creative Commons CC-BY-NC-ND license) (**b**) Angle of the intersection.

**Figure 10 nanomaterials-14-01113-f010:**
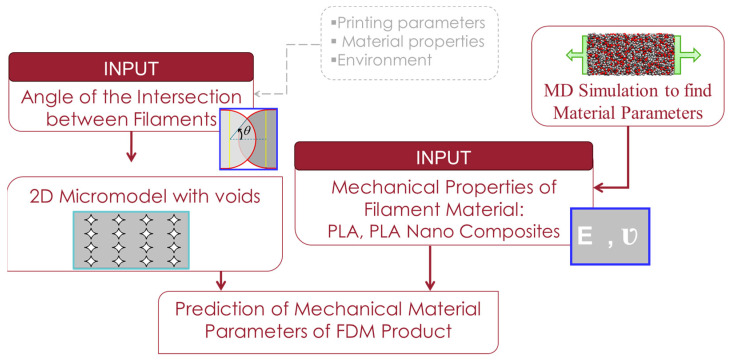
Outline of the methodology.

**Figure 11 nanomaterials-14-01113-f011:**
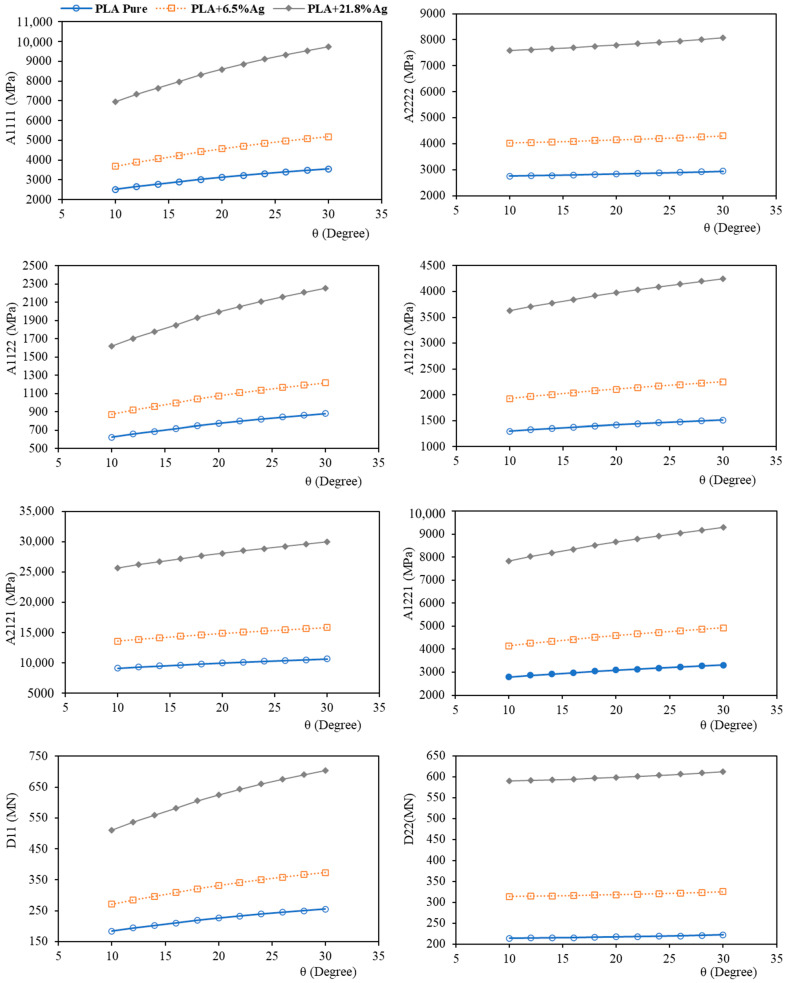
Effect of angle of intersection (θ) on mechanical properties in terms of equivalent micropolar parameters.

**Figure 12 nanomaterials-14-01113-f012:**
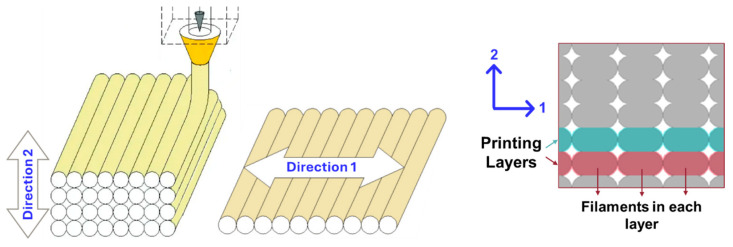
Different material orientations in the FDM-printed sample.

**Figure 13 nanomaterials-14-01113-f013:**
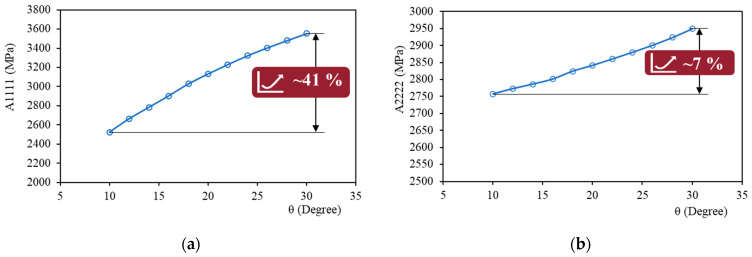
The impact of changing θ on equivalent mechanical parameters: (**a**) in direction 1 (high); (**b**) in direction 2 (low); and (**c**,**d**) on shear terms (medium).

**Figure 14 nanomaterials-14-01113-f014:**
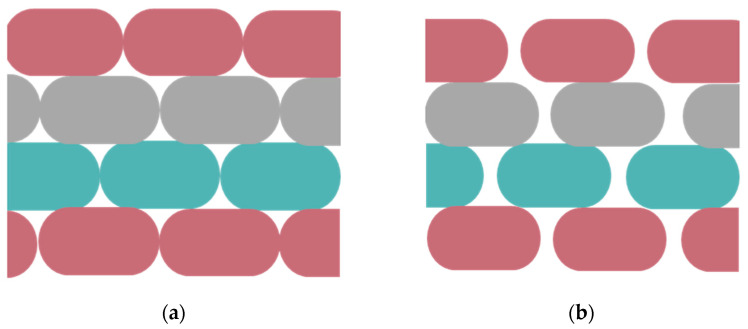
(**a**) FDM printing with non-matching void layout. (**b**) FDM printing with the presence of an air gap between layers.

**Table 1 nanomaterials-14-01113-t001:** Young’s moduli and Poisson’s ratios obtained from the MD simulation.

		Nanofiller Weight Fraction (%)	Young’s Modulus (GPa)	Poisson’s Ratio
1	Pure PLA	0.00	3.779	0.300
2	PLA reinforced by silver nanoparticles	6.50	5.558	0.281
3		21.8	10.461	0.276

**Table 2 nanomaterials-14-01113-t002:** Nomenclature of the thermal model variables suggested in [20] for FDM process.

Variable	Unit	Description
$h_{\mathrm{supp}}$	${W/m}^{2\;}{}^{\circ}C$	The thermal conductivity between the segments of the filament and the support material.
$T_{\mathrm{supp}}$	°C	Support temperature.
$a_{r\mathrm{supp}}$	-	=1 In the presence of contact between the filament element and the support; =0 otherwise.
$\lambda_{\mathrm{supp}}$	-	The fraction of the perimeter in contact with the support.
$T_{E}$	°C	Environment temperature.
$T_{extr}$	°C	Extrusion temperature.
$t_{r}(x)$	s	Time at which filament segment x of the filament r is deposited.
P	m	Perimeter of the cross-section.
A	$m^{2}$	Cross-sectional area.
$a_{ri}$	-	=1 if the filament segment r is in contact with another filament segment; =0 otherwise.
$h_{i}$	${W/m}^{2\;}{}^{\circ}C$	Heat transfer coefficient of contact *i.*
$\lambda_{i}$	-	Fraction of perimeter of filament in contact *i.*
C	${J/\mathrm{kg}\;}^{o}C$	Specific heat capacity.
ρ	${\mathrm{kg}/m}^{3}$	Density.
n	-	Number of physical contacts with an adjacent filament segment or support.

**Table 3 nanomaterials-14-01113-t003:** FDM process parameters for parametric studies.

Parameter		Value
	Sample size	L × L
$H_{0}$	Layer height	0.05 L
$W_{0}$	Initial flat width	0.05 L
θ	Intersection angle	10~30
	No. of filaments per layer	10
	No. of printing layers	20

## Data Availability

The data presented in this study are available on request.

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
