# Peer review of "A Hierarchical Nano to Micro Scale Modelling of 3D Printed Nano-Reinforced Polylactic Acid: Micropolar Modelling and Molecular Dynamics Simulation"

_nanomaterials, 2024, doi:10.3390/nano14131113_

Round 1
Reviewer 1 Report
Comments and Suggestions for Authors
The manuscript presents a comprehensive study on the multiscale modeling of 3D printed components made from PLA reinforced with silver nanoparticles, using FDM technology. The authors proposed a hierarchical model that spans nano to microscales to predict the mechanical properties of the 3D printed systems. At the nanoscale, molecular dynamics simulations are used to capture the behavior of the PLA matrix reinforced with nanoparticles; at the microscale, micropolar continuum theory is applied to include the effects of voids and other heterogeneities inherent in the FDM process. The topic is suitable for publication in Nanomaterials, but there are some technical comments that need further clarification.
· Can the authors elaborate on why silver nanoparticles were chosen and whether their effects on the mechanical properties of PLA are superior to other nanomaterials?
· The manuscript effectively links nanoscale and microscale phenomena, but it could better illustrate how these scales interact. Specifically, how do nanoscale changes impact microscale properties like mechanical strength and durability?
· Does the micropolar model account for the anisotropy in mechanical properties? How well does the model predict the directional differences in material behavior?
· It would be beneficial if the authors could compare the performance of the nano-reinforced PLA with other common composites used in FDM.
Author Response
Please kindly see the attachment for the responses to Reviewer's comments.

Reviewer 2 Report
Comments and Suggestions for Authors
The work presents an excellent and well-structured approach to modelling FDM components with microstructures. The choice of utilising the classical Cauchy continuum at the micro-level and the micropolar continuum at the macro-level is well-justified and highlights a deep understanding of the underlying mechanics.
Author Response

(The authors gave the same response as above.)

Reviewer 3 Report
Comments and Suggestions for Authors
Fused Deposition Modelling of components include complexities at nanoscale and microscale. Introduction of nano-additives into the FDM material enhances design flexibility and lead to significant improvement of the mechanical properties. However, design will be obviously more complex. The study presents multiscale modelling to predict the mechanical properties of FDM-manufactured components.
Remarks:
1. Fused Deposition Modelling is well described in introduction. However, please give short description an alternate suitable additive manufacturing techniques and explain why FDM selected.
2. Why nano silver particles are used for reinforcement of PLA? Short explanation can be added.
Conclusion: the study covers complex multiscale modelling: model development and application for prediction of mechanical parameters. Contribution of authors is obvious.
Author Response

(The authors gave the same response as above.)

Round 2
Reviewer 1 Report
Comments and Suggestions for Authors
All the comments have been clearly addressed and I would recommend the publication.